# SOX1 Functions as a Tumor Suppressor by Repressing HES1 in Lung Cancer

**DOI:** 10.3390/cancers15082207

**Published:** 2023-04-08

**Authors:** Shan-Yueh Chang, Ti-Hui Wu, Yu-Lueng Shih, Ying-Chieh Chen, Her-Young Su, Chih-Feng Chian, Ya-Wen Lin

**Affiliations:** 1Graduate Institute of Medical Sciences, National Defense Medical Center, Taipei 11490, Taiwan; leehornwok@gmail.com (S.-Y.C.); su108868@gmail.com (H.-Y.S.); 2Division of Pulmonary and Critical Care Medicine, Department of Internal Medicine, Tri-Service General Hospital, National Defense Medical Center, Taipei 11490, Taiwan; jeierchen@gmail.com (Y.-C.C.); sonice3982@gmail.com (C.-F.C.); 3Division of Thoracic Surgery, Department of Surgery, Tri-Service General Hospital, National Defense Medical Center, Taipei 11490, Taiwan; chestsurgerytsgh@gmail.com; 4Division of Gastroenterology, Department of Internal Medicine, Tri-Service General Hospital, National Defense Medical Center, Taipei 11490, Taiwan; albreb@ms28.hinet.net; 5Department and Graduate Institute of Microbiology and Immunology, National Defense Medical Center, Taipei 11490, Taiwan; 6Graduate Institute of Life Sciences, National Defense Medical Center, Taipei 11490, Taiwan

**Keywords:** SOX1, tumor suppressor, lung cancer, HES1

## Abstract

**Simple Summary:**

Lung cancer is the most common reason for cancer-related death, and patient survival is mainly dependent on tumor stage. The median overall survival of patients who are diagnosed with advanced/metastatic non-small-cell lung cancer (NSCLC) is still less than 3 years. The poor survival rates highlight the unmet need to elucidate the mechanism underlying lung cancer carcinogenesis to improve treatment responses and overall survival. The expression and function of SOX1 in the progression of lung cancer are still unclear. Through in vitro and in vivo experiments, we demonstrated that SOX1 repressed anchorage-independent growth, invasion, and metastasis. Interestingly, SOX1 performed its function by repressing hairy and enhancer of split 1 (HES1). Recent studies have shown that HES1, which is a helix–loop–helix transcription factor, performs important functions in stemness, metastasis, and drug resistance in cancer. These results suggest that SOX1 is a tumor suppressor that affects the carcinogenesis of lung cancer.

**Abstract:**

The development of lung cancer is a complex process that involves many genetic and epigenetic changes. Sex-determining region Y (SRY)-box (SOX) genes encode a family of proteins that are involved in the regulation of embryonic development and cell fate determination. SOX1 is hypermethylated in human cancers. However, the role of SOX1 in the development of lung cancer is unclear. We used quantitative methylation-specific polymerase chain reaction (MSP), quantitative reverse transcription polymerase chain reaction (RT–PCR) analysis, and web tools to confirm the frequent epigenetic silencing of SOX1 in lung cancer. Stable overexpression of SOX1 repressed cell proliferation, anchorage-independent growth, and invasion in vitro as well as cancer growth and metastasis in a xenograft mouse model. Knockdown of SOX1 by the withdrawal of doxycycline partly restored the malignant phenotype of inducible SOX1-expressing NSCLC cells. Next, we discovered the potential downstream pathways of SOX1 using RNA-seq analysis and identified HES1 as a direct target of SOX1 using chromatin immunoprecipitation (ChIP)-PCR. Furthermore, we performed phenotypic rescue experiments to prove that overexpression of HES1-FLAG in SOX1-expressing H1299 cells partly reversed the tumor-suppressive effect. Taken together, these data demonstrated that SOX1 acts as a tumor suppressor by directly inhibiting HES1 during the development of NSCLC.

## 1. Introduction

Lung cancer is the leading cause of cancer-related death globally, and there were approximately 1.8 million lung cancer-related deaths around the world in 2020 [1]. During the last decade, the treatment principles for non-small-cell lung cancer (NSCLC), which is a subtype of lung cancer that accounts for 85% of all cases, have been dramatically altered. This change mainly occurred due to the accessibility of biomarkers that allow the selection of patients for targeted and immunotherapy-based treatments [2]. With the use of second- and next-generation EGFR or ALK inhibitors, median survival is greater than 3 or 5 years, respectively [3,4]. Combinations of immunotherapy and chemotherapy are now widely used to treat patients who present with advanced or metastatic NSCLC and who do not harbor mutations in EGFR, ALK, or other drivers [5,6,7,8]. Initial clinical trials involving agents that target the PD-1/PD-L1 axis provided the chance of long-term survival for a small population with high PD-L1 expression. Despite these advances, the median overall survival of patients who are diagnosed with locally advanced/metastatic NSCLC with no detectable EGFR mutation and/or ALK translocation and with PD-L1 staining > 1% remains lower than 3 years [9]. These survival rates highlight the unmet need to elucidate the mechanism underlying lung cancer carcinogenesis in order to improve treatment responses and overall survival.

Abnormal activation of the Wnt/*β*-catenin pathway is important in the initiation, progression, metastasis, and drug sensitivity of lung cancer tumors [10,11,12,13,14,15,16,17,18]. Downregulation of Wnt antagonists through hypermethylation is common in human NSCLC cell lines and primary tumors, even though mutations in *β*-catenin or APC are relatively unusual in lung cancer [18]. The Wnt/*β*-catenin pathway is also crucial for the maintenance of cancer stem cells (CSCs) [16,17]. Sex-determining region Y (SRY)-box (SOX) family genes encode a group of proteins that share a similar DNA-binding high mobility group (HMG) domain [19], play an important role in the control of embryonic development, and are involved in the organization of cell fate [20]. Different members of the SOX family of proteins regulate diverse signaling pathways in cancers and can act as either tumor suppressor genes [21,22,23,24] or oncogenes [25,26,27]. Among the members of the SOX family, SOX1 is evolutionarily conserved in different species and promotes the neural differentiation of neuronal stem or progenitor cells [28,29,30]. SOX1 promotes neurogenesis by cooperating with *β*-catenin to repress *β*-catenin-mediated TCF/LEF signaling [29]. The function of the SOX family in carcinogenesis remains unclear. Some members of the SOX family have been shown to inhibit *β*-catenin activity by either promoting *β*-catenin degradation or interfering with the interaction between TCF/LEFs and *β*-catenin [31,32]. These results indicate that SOX family members commonly perform their functions via the regulation of Wnt signaling. Our previous data showed that SOX1 was hypermethylated in several cancers, including cervical cancer, ovarian cancer, and HCC [33,34,35]. Furthermore, we recently demonstrated that SOX1 acts as a tumor suppressor by interfering with the Wnt/*β*-catenin signaling pathway in HCC [35] and cervical cancer [29]. Moreover, SOX1 is frequently hypermethylated in NSCLC [36,37]. The expression and function of SOX1 in the development of lung cancer remain unclear.

In this study, our data confirmed that SOX1 was frequently downregulated via promoter hypermethylation. Moreover, overexpression of SOX1 significantly represses cancer growth and invasion in vitro and in vivo. Interestingly, SOX1 performs its function through hairy and enhancer of split 1 (HES1) but not the Wnt/*β*-catenin signaling pathway. Recent studies have shown that HES1, which is a helix–loop–helix transcription factor, performs important functions related to stemness, metastasis, and drug resistance in cancer [38,39,40,41,42]. These results demonstrate that SOX1 is a tumor suppressor during the carcinogenesis of lung cancer.

## 2. Materials and Methods

### 2.1. Clinical Tissue Samples

Tissue samples were obtained from the Taiwan Biobank and Biobank, Tri-Service General Hospital (TSGH), Taipei, Taiwan. The samples were collected from 70 patients with non-small-cell lung cancer whose diagnosis contained the histological subtype and staging. These specimens were collected during surgery, frozen directly in liquid nitrogen, and stored at −80 °C until laboratory analysis. The existence of malignant cells was approved by histological examination. The inclusion criteria of the enrolled patients were adults 20 years of age or older with histologically confirmed non-small-cell lung cancer. The de-identified demographic data were collected by the staff of the biobank anonymously to protect all subjects’ privacy. This study was approved by the TSGH institutional review board (TSGHIRB: C202005146).

### 2.2. Bioinformatics Analysis

Bioinformatics analysis was performed by using the UALCAN website (http://ualcan.path.uab.edu, accessed on 1 July 2019) [43] and the Shiny Methylation Analysis Resource Tool (SMART) App (http://www.bioinfo-zs.com/smartapp, accessed on 1 June 2020) [44]. We used the UALCAN to analyze the methylation levels of *SOX1* in the dataset from The Cancer Genome Atlas (TCGA). The SMART App was used to analyze the correlation between *SOX1* methylation and gene expression. DNA methylation beta value was the methylation index outputted for each probe site, which ranges between 0 and 1, representing the ratio of the intensity of the methylated signal to the intensity of the total signal.

### 2.3. Cell Lines

A total of six human lung cancer cell lines (H1299, H23, CL1-0, H1437, H358, and A549) and one immortalized lung cell line (BEAS-2B) were utilized in this study. They were obtained from Professor Yi-Ching Wang (National Cheng Kung University, Taiwan). Culture conditions and other components were employed as previously described [45].

### 2.4. DNA Methylation and Gene Expression Analysis

We extracted the genomic DNA of NSCLC cell lines and performed bisulfite conversion as previously described [46]. MSP, bisulfate sequencing, and Q-MSP were executed as previously described [33,35]. For Q-MSP, the DNA methylation levels were evaluated by determining the methylation index (MI) using the following formula: 100 × 2^−[(Cp of Gene) − (Cp of COL2A)]^. RNA extraction and reverse transcription polymerase chain reaction (RT-PCR) were used to analyze the gene expression, and the experiments were performed as previously described [35,45,46]. The primer sequences are listed in Appendix A [35]. Detailed descriptions are available in the Appendix A.

### 2.5. Plasmids and shRNA Clones

The full-length SOX1 open-reading frame (ORF) was cloned into the pcDNA3.1-V5-His-TOPO constitutive expression vector (termed pcDNA3.1-SOX1) or the inducible expression vector pT-REx-DEST31 (termed pT-REx-DEST-SOX1) as previously described [35]. The pLKO.1-shLacZ and HES1-shRNA were obtained from National RNAi Core Facility of Taiwan. The pENTER-HES1 plasmid was purchased from a biotech company (ViGene BioSciences, Rockville, MD, USA). The shRNA sequences were described in Appendix A.

### 2.6. Assays for Western Blot, Cell Viability, Anchorage-Independent Growth, and Invasion

Assays for Western blot, cell viability by MTS(3-(4,5-dimethylthiazol-2-yl)-5-(3-carboxymethoxyphenyl)-2-(4-sulfophenyl)-2H-tetrazolium, Inner Salt), anchorage-independent growth, and invasion were performed as described previously [47,48]. Detailed descriptions are accessible in the Appendix A. The following antibodies were used in Western blot: anti-SOX1 (R&D Systems, Minneapolis, MN, USA), anti-HES1 (Cell Signal, Danvers, MA, USA).

### 2.7. Immunofluorescence Staining

Assays for immunofluorescence staining were performed as described previously [35,48]. The following antibody was used in the immunofluorescence assay: anti-SOX1 (R&D Systems). Finally, DAPI was used for nuclei staining, and images were visualized using a fluorescence microscope (Leica, Wetzlar, Germany).

### 2.8. In Vivo Tumor Xenograft and Metastasis Model

Six-week-old nonobese diabetic severe-combined immunodeficiency (NOD-SCID) female mice were utilized in the tumorigenicity and metastasis analysis. All animal studies were approved by the Institutional Animal Care and Use Committee of the National Defense Medical Center (approval number: IACUC-22-037). Detailed tumor xenograft and metastasis analyses were performed as described previously [35,47,48].

### 2.9. RNA Sequencing Data Analysis

RNA-seq data generation and normalization were performed on an Illumina NovaSeq 500 platform. The data were analyzed as described in our past report [49]. Following, ontology analysis and KEGG pathway analysis were performed with DAVID (https://david.ncifcrf.gov/, accessed on 1 June 2021) [50].

### 2.10. Chromatin Immunoprecipitation Assay (ChIP Assay)

We used an EZ-Magna ChIP G Kit (Millipore, Burlington, MA, USA) to perform ChIP assays according to the manufacturer’s protocol. Detailed descriptions are accessible in the Appendix A [48].

### 2.11. Statistical Analysis

GraphPad Prism software (version 5; GraphPad Software, La Jolla, CA, USA) and SPSS software (IBM SPSS Statistics 21; Asia Analytics Taiwan, Taipei, Taiwan) were used to perform statistical analyses. All values are expressed as the mean ± SEM. The Mann–Whitney U test was used to determine differences between disease status and gene methylation levels. Student’s *t*-test and Mann–Whitney *U* test were used to compare relative RNA expression, colony number, cell invasion, and cell proliferation in the different stable transfectants. In vivo experiments were analyzed using the unpaired two-tailed *t*-test. In all cases, *p* < 0.05 was considered statistically significant.

## 3. Results

### 3.1. Promoter Hypermethylation of SOX1 in Lung Cancer Contributes to SOX1 Silencing/Downregulation

First, we examined the methylation levels of SOX1 in 473 lung adenocarcinoma (LUAD) and 370 lung squamous cell carcinoma (LUSC) samples from The Cancer Genome Atlas (TCGA) using the UALCAN website (http://ualcan.path.uab.edu, accessed on 1 July 2019). The average beta value was greater in the LUAD and LUSC groups than in the normal control group (*p* < 0.0001) (Figure 1A). A similar trend was observed in the LUSC group (*p* < 0.0001). Next, we investigated the SOX1 methylation level in 70 pairs of NSCLC tissues from the TSGH biobank and found that *SOX1* methylation was significantly higher in tumors than in the corresponding nontumor samples (Figure 1B). Moreover, we performed a receiver operating characteristic (ROC) curve analysis to distinguish between the NSCLC tissue samples and their nontumor counterparts (Figure 1C) and defined that the methylation frequency of *SOX1* was 42.86% under the best cutoff values (12.77). However, we did not have enough tissues to perform SOX1 mRNA analysis. We found that there is an inverse correlation between SOX1 methylation (two CpG sites) and mRNA expression using the SMART App (http://www.bioinfo-zs.com/smartapp, accessed on 1 June 2020) [44] to investigate the dataset from TCGA (Appendix A). Based on these data, downregulation of SOX1 might be due to the promoter hypermethylation of SOX1 in NSCLC. Then, we analyzed the methylation level and mRNA expression of SOX1 in lung cancer cell lines using quantitative MSP (qMSP) and quantitative RT-PCR (qRT–PCR) (Figure 1D). The results of bisulfate genomic sequencing confirmed that SOX1 was poorly methylated in nontumor lung tissues and CL1-0 cells and highly methylated in H1299, H23, A549, H358, and H1437 cells (Appendix A). To confirm whether promoter methylation is correlated to the regulation of SOX1, two NSCLC cell lines (H1299 and H23) with no detectable SOX1 expression were treated with 5-AZA-2′-deoxycytidine (DAC). The data showed reduced methylation of SOX1 and upregulation of SOX1 mRNA in all the tested cell lines (Figure 1E), further revealing that the transcriptional silencing of SOX1 was due to promoter hypermethylation. Moreover, SOX1 was re-expressed in CL1-0 cells after they were treated with DAC and trichostatin A (TSA), suggesting that histone modification contributes to SOX1 silencing in CL1-0 cells (Appendix A).

### 3.2. Ectopic Expression of SOX1 Suppresses Lung Cancer Growth and Invasion

To determine whether SOX1 exerts a tumor-suppressive function, we investigated the growth-suppressive effect of SOX1 by restoring SOX1 expression in H1299, H23, and CL1-0 cells, which display no detectable SOX1 expression (Figure 1F). Stable expression of SOX1 in H1299, H23, and CL1-0 cells was verified by Western blotting analysis (Figure 2A). Overexpression of SOX1 in H1299 and CL1-0 cells significantly decreased cancer cell growth (Figure 2B). As shown in Figure 2C,D, restoration of SOX1 expression in H1299, H23, and CL1-0 cells significantly repressed anchorage-independent growth (AIG) and invasion.

### 3.3. Restoration of SOX1 Expression Inhibits Tumor Growth and Metastasis in NOD/SCID Mice

To evaluate the influence of SOX1 expression on tumor growth, we subcutaneously injected tumor cells expressing SOX1 into NOD/SCID mice. The subcutaneous tumor growth of H1299 cells that were stably transfected with empty vector or SOX1 into NOD/SCID mice is indicated in Figure 3A. The mean tumor volume was significantly reduced in the SOX1-transfected NOD/SCID mice compared to that in the vector control mice (*p* < 0.001). After 5 weeks, the tumors were removed and weighed. The mean tumor weight was significantly lower in the SOX1-transfected NOD/SCID mice than in the vector control mice (*p* < 0.001) (Figure 3B). The SOX1 expression in tumors from the SOX1-transfected and vector control groups was measured by Western blotting (Figure 3C). To further validate the function of SOX1 in metastasis in vivo, we injected SOX1-H1299 cells or control vector-H1299 cells into mice through the tail vein (Figure 3D–F). Six weeks after injection, there were fewer metastatic lung nodules in the SOX1 group than in the control group (Figure 3D–F). Our data further verified that the restoration of SOX1 expression represses tumor growth and metastasis in xenograft models.

### 3.4. SOX1 Suppresses Lung Cancer Cell Growth and Invasion in an Inducible Expression System

To further confirm the tumor suppressor function of SOX1, we applied a Tet-on inducible system to manipulate SOX1 expression. First, H1299 cells were treated with doxycycline (DOX) for 7 days to induce SOX1 expression, and then, knockdown of SOX1 expression was executed by removing DOX for another 7 days. Additionally, another group of cells was treated with DOX alone for 7 days. The SOX1 levels in both groups were confirmed by Western blotting analysis (Figure 4A). The detailed manipulation of SOX1 expression was described in our previous paper [35], and MTS and AIG assays were performed on schedule (Appendix A). The results indicated that inducible SOX1 expression significantly repressed cell growth compared with the control group, whereas knockdown of SOX1 expression by doxycycline withdrawal reversed the effects of inducible SOX1 expression on the growth of H1299 cells (Figure 4B). Furthermore, inducible SOX1 expression significantly suppressed colony formation and cancer invasion compared with the control group. Knockdown of SOX1 expression by doxycycline withdrawal significantly reversed colony formation and cell invasion of lung cancer cells (Figure 4C,D). These data were consistent with constitutively stable SOX1-expressing cell lines. In addition, we investigated the cellular localization of SOX1 using fluorescence microscopy. As illustrated in Figure 4E, the merged images indicated that SOX1 protein was localized in the nuclei of H1299 cells. Overall, these data confirm that SOX1 inhibits cell growth and invasion in an inducible system and that inhibition of SOX1 partially reverses the malignant phenotype.

### 3.5. Signaling Pathways Affected by Inducible Expression of SOX1

In our previous study, SOX1 suppressed *β*-catenin-mediated TCF/LEF signaling via interaction with *β*-catenin in HCC [35]. To further investigate whether SOX1 could inhibit the Wnt signaling pathway in lung cancer, we applied a Wnt/TCF-responsive luciferase reporter assay. The results indicated that overexpression of SOX1 did not significantly inhibit the TCF transcriptional activity compared to control/vector cells (Appendix A). Furthermore, there was no considerable change in the nuclear accumulation of *β*-catenin in the SOX1-overexpressing group compared with the control group (Appendix A). We next investigated whether the influence of SOX1 on invasion relates to the regulation of epithelial–mesenchymal transition (EMT) [51]. However, SOX1 overexpression did not significantly change the expression of the EMT markers N-cadherin, vimentin, Snail, Twist, and Slug at the mRNA level, as indicated in Appendix A.

To elucidate the molecular mechanism by which SOX1 controls cell migration and invasion, we executed RNA-seq analysis in H1299 cells after Dox induction or removal (Appendix A). This Venn diagram shows that SOX1 either directly or indirectly controls the expression of 53 genes (Figure 5A, Appendix A). In accordance with the tumor suppressor function of SOX1, Gene Ontology analysis and KEGG pathway analysis (Figure 5B) using DAVID software (https://david.ncifcrf.gov/, accessed on 1 June 2021) [50] discovered enrichment in pathways related to cell proliferation and cell migration, such as the pathway in cancers (*p* = 3.40 × 10^−2^) and pathways related to focal adhesion (*p* = 7.5 × 10^−4^). Quantitative RT–PCR analysis was used to verify that SOX1 controlled the expression of genes related to pathways in cancer, such as LAMB3, FN1, and HES1, and the focal adhesion pathway, including LAMB3 and FN1. Among these genes, we discovered and confirmed one putative target gene, namely HES1, that plays a role in the regulation of pathways in cancers (Appendix A). Indeed, overexpression of SOX1 significantly repressed the expression of HES1 at both the mRNA and protein levels in two lung cancer cell lines (Figure 5C). Our results provide a proof of concept and suggest that SOX1 might perform its tumor-suppressive function by inhibiting the expression of HES1.

### 3.6. HES1 Is a Direct Target of SOX1

We applied inducible SOX1-expressing H1299 cells to further confirm the influence of SOX1 on HES1 expression. HES1 protein and mRNA expression significantly decreased with the induction of SOX1 expression after DOX treatment. Then, the expression of HES1 was partially restored when SOX1 was knocked down in H1299 cells by the withdrawal of DOX (Figure 6A). Next, we investigated whether SOX1 directly regulates the transcription of HES1 using the Eukaryotic Promoter Database (EPD) (http://epd.vital-it.ch, accessed on 1 June 2021). We discovered one putative SOX1 binding site in the HES1 promoter, which is located at −1276 bp from the transcriptional start site (TSS) (Figure 6B). ChIP assays utilizing an antibody against SOX1 were then performed to investigate whether SOX1 can directly regulate HES1 expression. We designed primers to amplify this region between −1396 bp and −1181 bp from the TSS. As shown in Figure 6C, SOX1 binds to the predicted region of the HES1 promoter. Additionally, and consistent with increased SOX1 binding, we also observed a significant increase in methylation of H3K27 (Me3H3K27) in the HES1 promoter (Figure 6C). These data suggest that SOX directly binds to the HES1 promoter and reduces its expression.

### 3.7. SOX1 Acts as a Tumor Suppressor by Repressing HES1 in Lung Cancer

To further investigate the role of HES1 expression in the carcinogenesis of lung cancer, we knocked down HES1 with HES1-shRNA and performed colony formation and invasion assays in H1299 cells. After the knockdown of HES1 in H1299 cells, the malignant phenotypes were significantly repressed (Figure 7A,B, Appendix A). Our findings further suggest that SOX1 inhibits colony formation and cell invasion by partly regulating the expression of HES1. To confirm this, we performed phenotypic rescue experiments. Overexpression of HES1-FLAG in SOX1-expressing H1299 cells partly reversed the tumor-suppressive effect of SOX1 expression compared with the control (Figure 7C,D). Similar phenomena were verified in H23 cells (Appendix A). Collectively, these data suggest that the downregulation of HES1 contributes to the suppressive effect of SOX1 in lung cancer.

## 4. Discussion

Our study elucidated the function of the transcription factor SOX1 in the regulation of colony formation and invasion in NSCLC cells, implying its function as a tumor suppressor (TSG). These data provide in vitro and in vivo evidence that demonstrates that SOX1 suppresses tumorigenicity and cancer metastasis. Interestingly, downregulation of HES1 contributes to the suppressive effect of SOX1 on colony formation and cell invasion in lung cancer. SOX1 performs its TSG function by directly repressing HES1.

The hypermethylation of SOX1 was confirmed in TSGH tissue samples using a Q-MSP assay. We proved that SOX1 methylation inversely correlated with its mRNA expression levels in lung cancer cell lines. Furthermore, we analyzed SOX1 promoter methylation levels in the TCGA database and confirmed that significant SOX1 hypermethylation was common in both the LUAD and LUSC groups. Kontic et al. reported that 5-year overall survival was significantly shorter in patients with resected NSCLC and concomitant SOX1 hypermethylation [36]. However, due to limited clinical information and sample numbers, we did not analyze the correlation between SOX1 methylation and clinical parameters. Because lung or NSCLC cell lines expressing SOX1 were not available, we used DOX withdrawal from an inducible SOX1 expression system to mimic the knockdown strategy. Overexpression of SOX1 in an inducible system significantly suppressed the malignant phenotype of a lung cancer cell line, whereas knockdown of SOX1 expression partially restored cancer cell invasion and anchorage-independent growth.

*SOX1* is hypermethylated in cervical cancer, ovarian cancer, and HCC according to our previous studies [33,34,35]. It is well known that *SOX1* represses *β*-catenin-mediated TCF/LEF signaling by interaction with *β*-catenin in hepatocellular carcinoma and nasopharyngeal carcinoma (NPC) [35,52]. Here, we demonstrated that SOX1 did not significantly affect the Wnt/*β*-catenin pathway or EMT in NSCLC. RNA-seq technology was used in inducible SOX1 H1299 cells after DOX treatment and DOX withdrawal to compare differentially expressed genes (DEGs). After qPCR, Western blotting, and putative SOX1 binding site prediction, we hypothesized that HES1 was the direct downstream target of SOX1. Our ChIP data proved that SOX1 directly binds to the promoter region of HES1. Phenotypic rescue experiments further suggested that downregulation of HES1 contributes to the suppressive effect of SOX1 on colony formation and cell invasion in lung cancer.

According to a literature review, overexpression of SOX1 in cultured neural progenitor cells could induce neural cell differentiation. SOX1 promotes neuronal lineage commitment by binding directly to the HES1 promoter, followed by suppressing the transcription of HES1 and attenuating the Notch pathway [53]. Hairy and enhancer of split 1 (HES1) performs important functions in diverse physiological processes, such as cell cycle arrest, apoptosis, cellular differentiation, and self-renewal ability. In cancer research, HES1 has been reported to play crucial roles in the maintenance of cancer stem cell self-renewal, induction of the EMT process, cancer metastasis, and antagonism of drug-induced apoptosis [38,39,40,41]. Furthermore, HES1 expression was significantly correlated with poor overall survival of NSCLC in a meta-analysis [54]. Li and his colleagues showed that FOXP3 regulated the Notch1/HES1 pathway to enhance the malignant phenotype, including invasion and metastasis, of NSCLC cells [39]. HES1 lies at the crossroads of multiple signaling pathways, including the Notch, EGFR, FGFR, Wnt/*β*-catenin, and Hedgehog signaling pathways, and may provide a new strategy for anticancer therapy [41]. Notch1 and HES1 were upregulated after treatment with a tyrosine kinase inhibitor (TKI) in approximately half of EGFR-mutated NSCLC tumors [42]. A γ-secretase inhibitor (GSI) inhibits Notch target gene expression, which is followed by inhibition of malignant phenotypes, without specificity, and therefore causes more gastrointestinal toxicities due to the rapid differentiation of intestinal progenitor cells into goblet cells. The combination of osimertinib (TKI) and GSI may be potentially beneficial for patients with EGFR-mutant NSCLC because NOTCH1 and Notch target genes are increased in osimertinib drug-tolerant persister (DTP) cells [42]. If we can target molecules downstream of the Notch pathway, such as HES1, we may achieve similar anticancer effects and observe less toxicity since many other Notch target genes will not be affected. A combination strategy with EGFR TKIs and HES1 inhibitors may be an alternative treatment strategy to delay the resistance to gefitinib caused by the T790M mutation in NSCLC. Further exploration of whether SOX1 suppresses HES1 in EGFR-mutant NSCLC cells and the role of SOX1 in TKI resistance is warranted. The reason that we did not combine ChIP-seq and RNA-seq to explore the downstream target genes of SOX1 is the lack of a good anti-SOX1 antibody. Our results suggest that SOX1 might perform its tumor-suppressive function by inhibiting the expression of HES1. Comprehensive analysis of the networks regulated by SOX1 requires further investigation. In this study, we demonstrated that the SOX1/HES1 axis contributes to cancer growth and invasion in lung cancer.

## 5. Conclusions

Taken together, these results showed that SOX1 is frequently downregulated by promoter hypermethylation in non-small-cell lung cancer, which may lead to aberrant activation of HES1. Restoration of SOX1 expression repressed HES1 via the direct binding of SOX1 to the HES1 promoter region and inhibited the malignant phenotype of lung cancer cells. These findings indicate that SOX1 may serve as an important tumor suppressor gene by suppressing HES1 during the development of NSCLC.

## Figures and Tables

**Figure 1 cancers-15-02207-f001:**
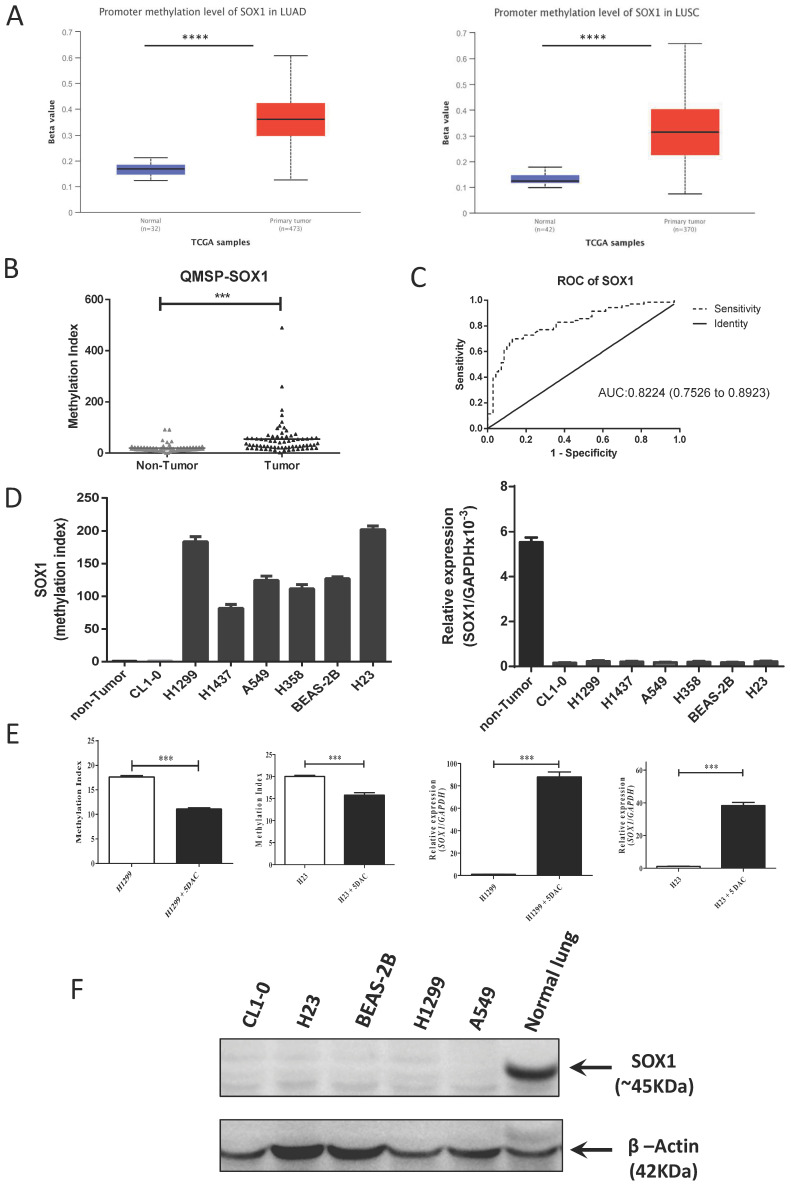
Promoter hypermethylation of SOX1 in lung cancer contributes to SOX1 silencing/downregulation. (**A**) DNA methylation array data for SOX1 in 42 samples from healthy individuals, 473 samples from lung adenocarcinoma (LUAD) patients, and 370 samples from lung squamous cell carcinoma (LUSC) patients from UALCAN (http://ualcan.path.uab.edu/, accessed on 1 July 2019) are shown. The results are shown as average (AVG) beta values for the probes. Black lines show the mean AVG beta value. The *p* values for SOX1 methylation levels between the groups (normal versus tumor) were analyzed using the Mann–Whitney *U* test. **** *p* < 0.0001. (**B**) The DNA methylation levels of SOX1 were examined in 70 paired lung cancer tissues and their adjacent nontumor tissues (NT) by quantitative methylation-specific PCR (Q-MSP). The results are presented as the difference in the methylation index (MI). The black lines show the mean of the MI. The *p* value for methylation levels among the groups was analyzed using the Wilcoxon signed-rank test. (**C**) Receiver operating characteristic (ROC) curves were created to determine the optimal cutoff point of SOX1 methylation for discriminating lung cancer and nontumor tissues. (**D**) Quantitative DNA methylation levels of SOX1 in one immortalized human lung epithelial cell line and six NSCLC cell lines were analyzed by Q-MSP. Gene expression levels of SOX1 were analyzed by quantitative RT–PCR. GAPDH was the internal reference control. (**E**) Quantitative DNA methylation levels of SOX1 in H23 and H1299 cells treated with 1 µM 5′-aza-2′-deoxycytidine (5DAC) or untreated were analyzed by Q-MSP. The results are presented as changes in the MI. Gene expression levels of SOX1 and the internal reference GAPDH in H23 and H1299 cells treated with 1 µM 5DAC or untreated were analyzed by quantitative RT–PCR. *** *p* < 0.001, (Student’s *t*-test). (**F**) SOX1 protein expression in human lung cancer cell lines was determined by Western blotting using an anti-SOX1 antibody. *β*-Actin was an internal control. The uncropped blots are shown in Appendix A.

**Figure 2 cancers-15-02207-f002:**
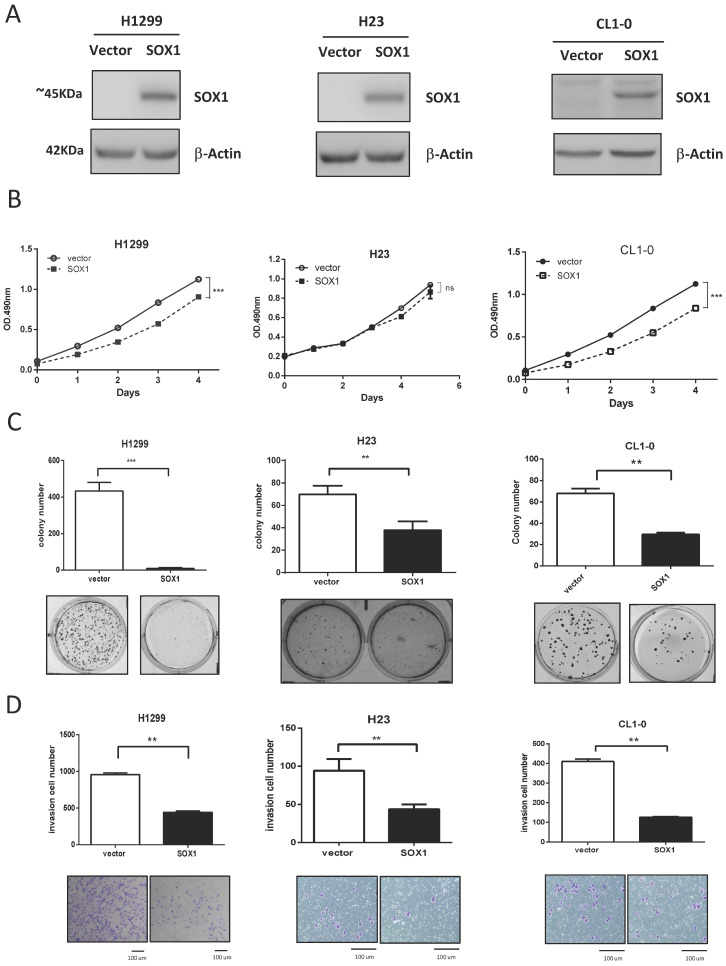
Ectopic expression of SOX1 suppresses lung cancer growth and invasion. (**A**) Stable overexpression of SOX1 was established in different lung cancer cell lines and shown by Western blotting. (**B**,**C**) Cell proliferation (MTS) assays (**B**) and anchorage-independent growth (AIG) assays (**C**) were performed in H1299, H23, and CL1-0 cells with SOX1 expression. (**D**) Matrigel invasion assays were performed in H1299, H23, and CL1-0 cells expressing SOX1. The data are expressed as the mean ± SE. Significant differences were analyzed using the Mann–Whitney *U* test. ** *p* < 0.01, *** *p* < 0.001, ns: not significant. The uncropped blots are shown in Appendix A.

**Figure 3 cancers-15-02207-f003:**
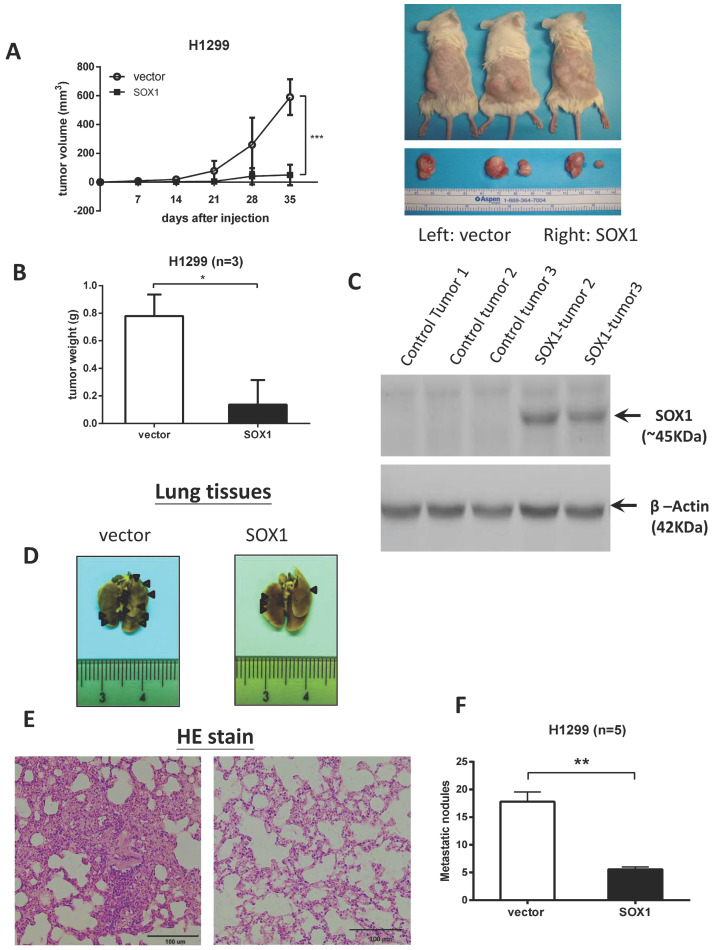
Restoration of SOX1 inhibits tumor growth and metastasis in xenograft mouse models. (**A**) SOX1-H1299 cells or control vector-H1299 cells were subcutaneously injected into the left and right flanks of NOD/SCID mice. The tumor growth curve of SOX1-expressing cells was compared with that of vector-alone cells. (**B**) Tumor weights from the SOX1 and vector alone groups. (**C**) SOX1 expression in the tumors from mouse xenografts was confirmed by Western blotting. (**D**) SOX1-H1299 cells or control vector-H1299 cells were injected into mice through the tail vein. After six weeks, lung tissues were excised from the mice; the arrows indicate lung nodules. (**E**) Representative images of hematoxylin and eosin (H&E) staining of lungs (original magnification, ×200) from the mice. (**F**) Lung nodule number. The data are expressed as the mean ± SE. Significant differences were analyzed using the unpaired two-tailed *t*-test. * *p* < 0.05, ** *p* < 0.01, *** *p* < 0.001. The uncropped blots are shown in Appendix A.

**Figure 4 cancers-15-02207-f004:**
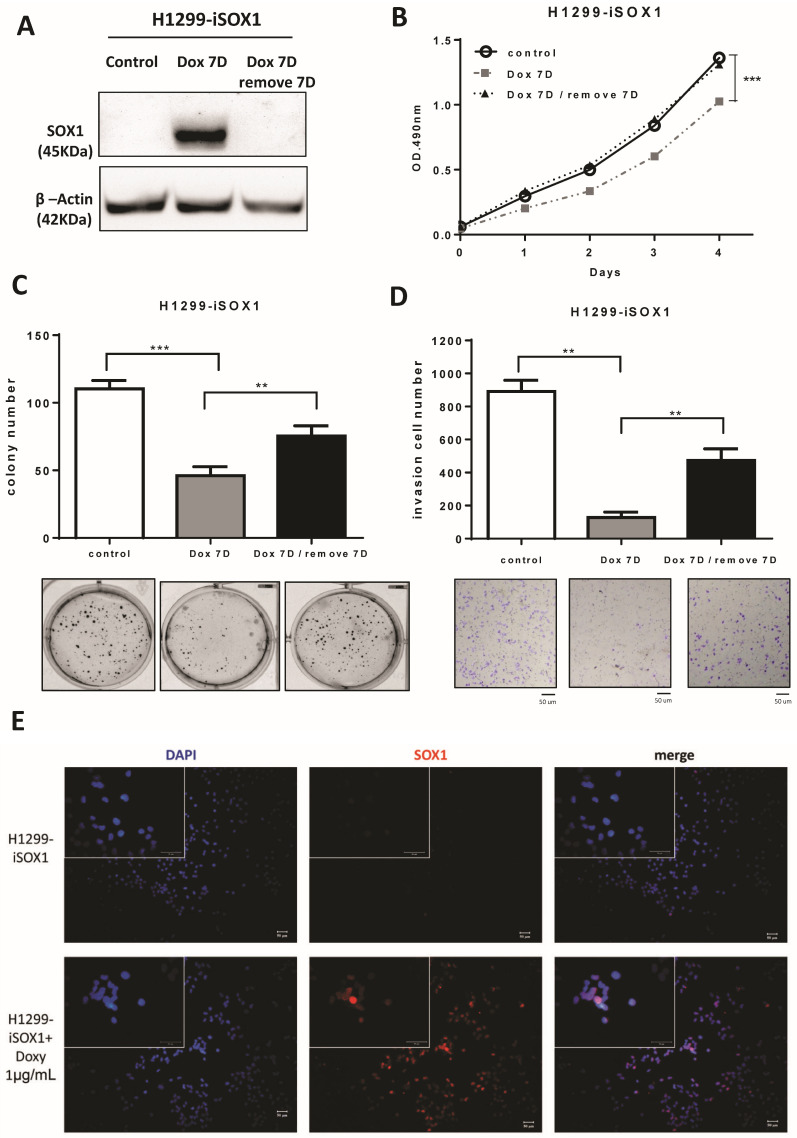
SOX1 represses lung cancer cell growth in an inducible expression system, and knockdown of SOX1 by DOX withdrawal partly reverses malignant phenotypes. (**A**) DOX (1 µg/mL)-inducible SOX1 expression was established in H1299 cells, and SOX1 expression after treatment with DOX for 7 days or withdrawal of DOX treatment for another 7 days was verified by Western blotting analysis. The detailed manipulations of SOX1 expression are illustrated in the Appendix A. Then, MTS and AIG assays were performed according to the determined schedule. (**B**) MTS assays were used to examine the effect of SOX1 on the proliferation of H1299 cells. (**C**) Colony formation assay was applied to analyze the effect of SOX1 on cell growth. (**D**) Matrigel invasion assay was performed in H1299 cells treated with DOX (1 µg/mL) or after the withdrawal of DOX treatment. The data are displayed as the mean ± SE. Significant changes were determined using the Mann–Whitney *U* test. ** *p* < 0.01, *** *p* < 0.001. (**E**) The merged images indicate colocalization of SOX1 and nuclei by fluorescence microscopy. The SOX1 proteins were stained with Alexa Fluor 594-conjugated secondary antibodies (red). Cell nuclear DNA was marked with DAPI (blue signal). The uncropped blots are shown in Appendix A.

**Figure 5 cancers-15-02207-f005:**
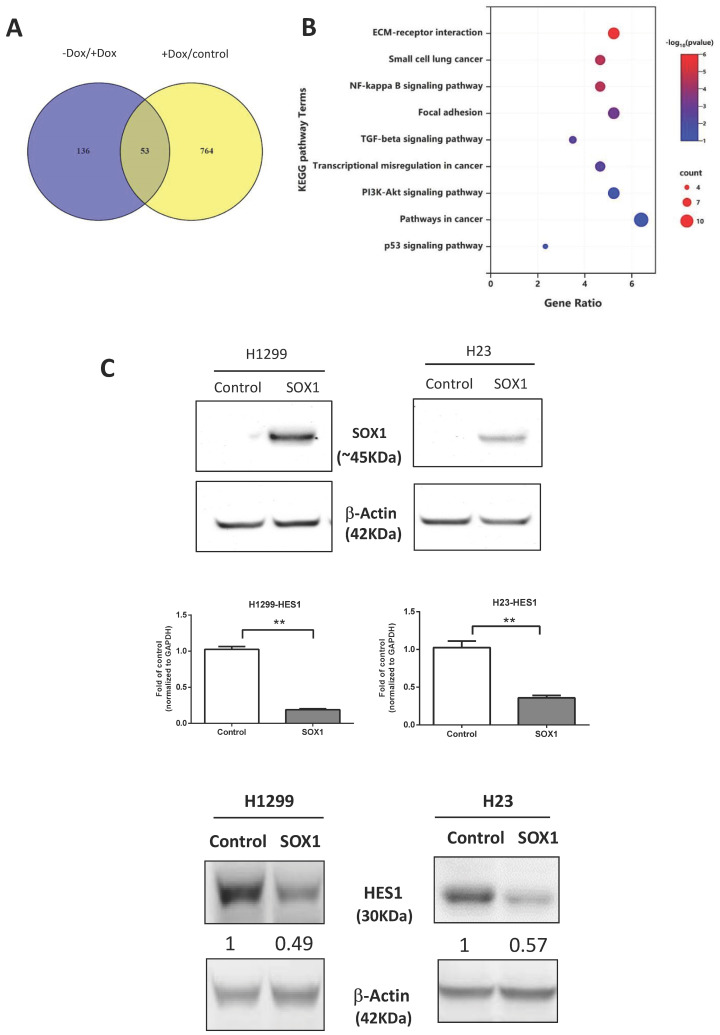
The KEGG signaling pathway affected by inducible expression of SOX1. (**A**) Venn diagram displaying the overlap between genes controlled by inducible SOX1 in H1299 cells. (**B**) Gene Ontology analysis of the most representative pathways controlled by SOX1. The analysis was accomplished with DAVID (https://david.ncifcrf.gov/, accessed on 1 June 2021). (**C**) Expression of SOX1 resulted in a reduction in the mRNA and protein levels of HES1 in H1299 and H23 lung cancer cells. The mRNA levels of the HES1 were analyzed by quantitative RT–PCR. The data were normalized to the housekeeping gene GAPDH and are shown as relative to the vector control. The data are expressed as the mean ± SE. ** *p* < 0.01. Protein levels were analyzed by Western blotting. The numbers in the Western blots indicate the ratios of HES1 expression to that of the internal control. The uncropped blots are shown in Appendix A.

**Figure 6 cancers-15-02207-f006:**
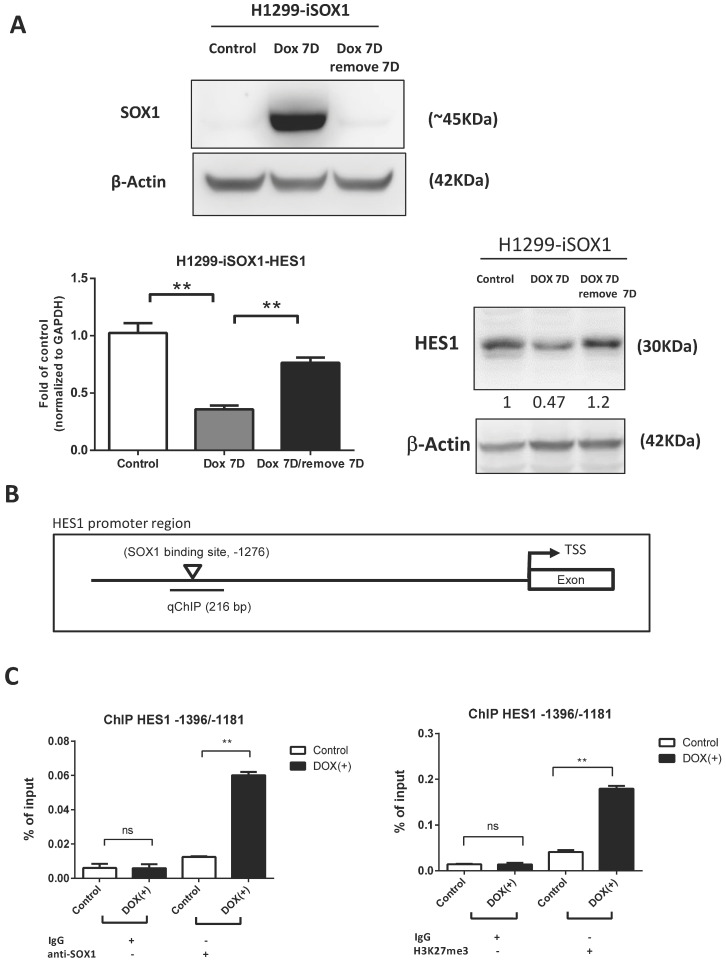
Hes1 is a direct target of SOX1. (**A**) SOX1 expression levels after treatment with DOX or withdrawal of DOX treatment were determined by Western blotting analysis. The mRNA and protein levels of HES1 in inducible SOX1-expressing H1299 lung cancer cells were determined via quantitative RT–PCR and Western blotting. *β*-Actin was used as the internal control for Western blotting. mRNA levels of the indicated genes were normalized to those of the housekeeping gene GAPDH and are shown as relative to the internal control. The data are shown as the mean ± SE of three independent experiments. The numbers in the Western blots indicate the ratios of HES1 expression to that of the internal control. (**B**) A diagram of the HES1 promoter with putative SOX1 binding sites. One predicted Sox1 binding site in the HES1 promoter was located at −1276 bp from the transcriptional start site (TSS). (**C**) ChIP assays were performed in SOX1-inducible H1299 cells. IgG antibody was a negative control. Protein–chromatin complexes were precipitated using SOX1 antibody and H3K27 (Me3H3K27) antibody, and the purified DNA was amplified by qPCR. Statistical significance was determined using the Mann–Whitney *U* test. ** *p* < 0.01, ns: not significant. The uncropped blots are shown in Appendix A.

**Figure 7 cancers-15-02207-f007:**
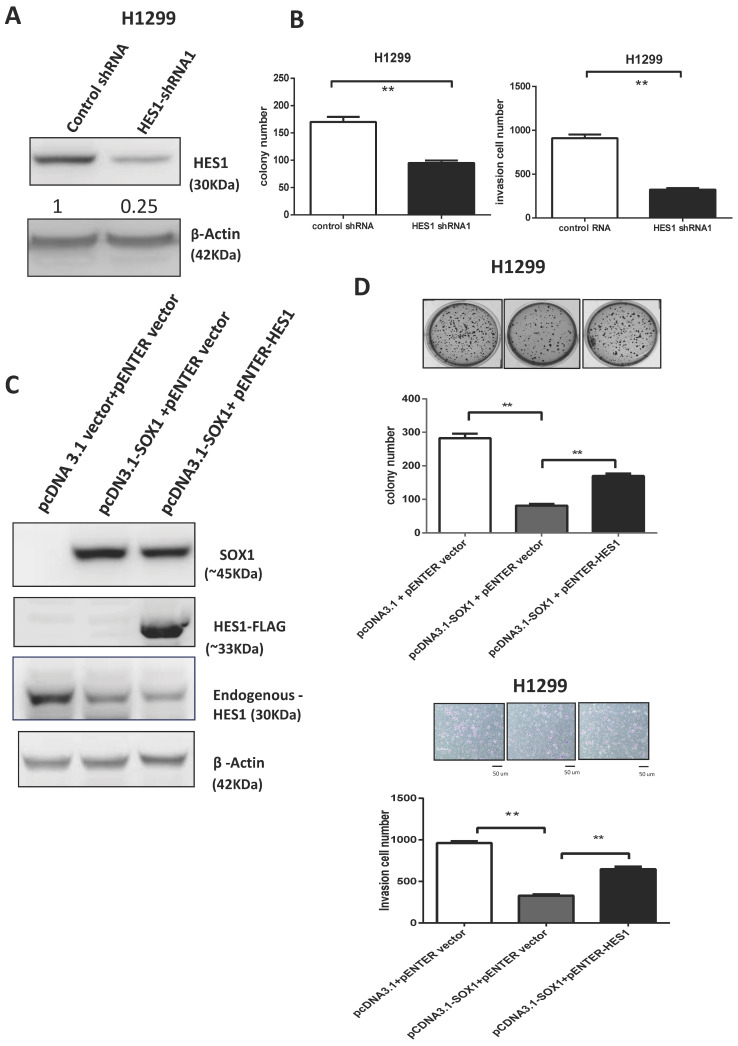
SOX1 acts as a tumor suppressor by repressing HES1 in lung cancer. (**A**) The expression of SOX1 in H1299 cells transfected with control shRNA (shCtrl) or HES1 shRNA was investigated by Western blotting analysis. The numbers in the Western blots indicate the ratios of HES1 expression to that of the internal control. (**B**) Colony formation and invasion assays were performed in H1299 lung cancer cells. (**C**) H1299 cells were transfected with the specified combination of vectors, and SOX1, HES1, and HES1-FLAG were analyzed by Western blotting analysis. (**D**) Colony formation and Matrigel invasion assays were applied to analyze the effects on anchorage-independent growth and cancer invasion. The data are presented as the mean ± SE. Statistical significance was calculated with the Mann–Whitney *U* test. ** *p* < 0.01. The uncropped blots are shown in Appendix A.

## Data Availability

The authors declare that all data supporting the findings of this study are available in the main text or the Appendix A.

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
