# Peer review of "SOX1 Functions as a Tumor Suppressor by Repressing HES1 in Lung Cancer"

_cancers, 2023, doi:10.3390/cancers15082207_

Round 1

Reviewer 1 Report

minor concerns: Authors should indicate in the methods section (or in the supplementary section) the number of cells seeded for the MTS assays. They should provide the reader with the definition of "beta-value" and define the terms of the equation used to calculate the MI. Molecular weights should be indicated on all figures with WB images and WB replicates should be shown in the supplementary.

Reviewer 2 Report

In this report, Chang and coworkers describe Sox1 as a tumor suppressor of NSCLC functioning via direct transcriptional inhibition of Hes1 expression. Their study closely follows the scheme of their own earlier report describing Sox1 as a tumor suppressor of hepatocarcinoma cells. The only significant difference is that  inhibitory targets are different. Overall, the study is fairly straight forward, and results are believable. There are however room for improvement.

Major points:

1. There is an earlier classic study about the role of Sox1 (Nat Neurosci 6(11):1162–1168) in which Sox1 has been described as a transcriptional activator not an inhibitor. This study apparently proposes the opposite. Does Sox1 really functions through the proposed binding site as an inhibitor? I think the provided data in this manuscript are minimally consistent but not sufficiently convincing.  I would like to propose reporter assays assessing the significance of the proposed binding sites along with domain swapping assay described in the aforementioned report. It may also help to assess the conservation of the binding site across the species.

2. Studies using cell lines should utilize as many cell lines as possible to strengthen the validity of results. Some assays such as Dox induction may be difficult but others such as those in figure 7 should be bolstered with use of additional cell lines.

Minor points:

1. In line 64, 'PD-L1>1%' is found. What does this mean? It should be clarified.

2. In Figure 1E, there should be protein blots to support the results.

3. Tet-on system should be described in the Materials and Methods section.

Round 2

Reviewer 2 Report

I am somewhat disappointed that the authors chose not to do further substantial work but are just trying to explain and justify their positions. While their report is believable as is, I wish the authors find that building up a more convincing case with additional data is a  worthwhile effort.